# IDRNet: Intervention-Driven Relation Network for Semantic Segmentation

**Zhenchao Jin[1], Xiaowei Hu[2], Lingting Zhu[1], Luchuan Song[4], Li Yuan[3], Lequan Yu[1]**
[1]The University of Hong Kong    [2]Shanghai AI Laboratory
[3]Peking University    [4]University of Rochester
{blwx96@connect., ltzhu99@connect., lqyu@}hku.hk
huxiaowei@pjlab.org.cn
yuanli-ece@pku.edu.cn
lsong11@ur.rochester.edu

## Abstract

Co-occurrent visual patterns suggest that pixel relation modeling facilitates dense prediction tasks, which inspires the development of numerous context modeling paradigms, *e.g.*, multi-scale-driven and similarity-driven context schemes. Despite the impressive results, these existing paradigms often suffer from inadequate or ineffective contextual information aggregation due to reliance on large amounts of predetermined priors. To alleviate the issues, we propose a novel **I**ntervention-**D**riven **R**elation **Net**work (**IDRNet**), which leverages a deletion diagnostics procedure to guide the modeling of contextual relations among different pixels. Specifically, we first group pixel-level representations into semantic-level representations with the guidance of pseudo labels and further improve the distinguishability of the grouped representations with a feature enhancement module. Next, a deletion diagnostics procedure is conducted to model relations of these semantic-level representations via perceiving the network outputs and the extracted relations are utilized to guide the semantic-level representations to interact with each other. Finally, the interacted representations are utilized to augment original pixel-level representations for final predictions. Extensive experiments are conducted to validate the effectiveness of IDRNet quantitatively and qualitatively. Notably, our intervention-driven context scheme brings consistent performance improvements to state-of-the-art segmentation frameworks and achieves competitive results on popular benchmark datasets, including ADE20K, COCO-Stuff, PASCAL-Context, LIP, and Cityscapes. Code is available at `https://github.com/SegmentationBLWX/sssegmentation`.

Table 1: Performance comparison between existing context schemes and our IDRNet.

| Benchmark | Multi-Scale-Driven | | | Similarity-Driven | | | Intervention-Driven | |
|---|---|---|---|---|---|---|---|---|
| | *ASPP* [8] | *PPM* [83] | *UPerNet* [70] | *Non-Local* [69] | *Ann* [89] | *OCR* [79] | *IDRNet* | *IDRNet+* |
| ADE20K [87] | 43.19 | 42.64 | 43.02 | 42.15 | 41.75 | 42.47 | 43.61 | **44.84** |
| Cityscapes [17] | 79.62 | 79.05 | 79.08 | 78.34 | 78.36 | 79.40 | 79.91 | **80.81** |

## 1 Introduction

Semantic image segmentation, which aims at assigning a semantic category to each pixel, is one of the most fundamental research areas in computer vision. Recent years have witnessed great progress in semantic segmentation using deep convolutional neural networks [44] and transformers [64]. The state-of-the-art semantic segmentation approaches [8, 12, 30, 39, 40, 50, 67, 79, 86] mostly follow the encoder-decoder paradigm [51] where the decoder is defined by replacing the fully-connected layers in classification tasks with some convolution layers.

37th Conference on Neural Information Processing Systems (NeurIPS 2023).

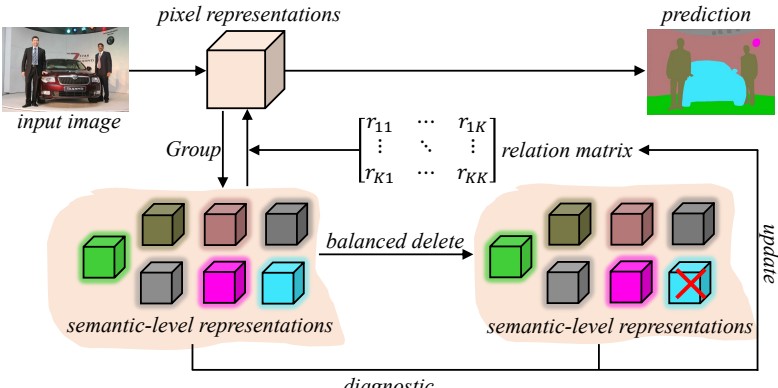

Figure 1: Diagram of our intervention-driven relation network. Deletion diagnostics is leveraged to build relations between semantic-level representations. With the built relation matrix and semantic-level representations, pixel representations can be augmented for pixel prediction.

Inspired by co-occurrent visual patterns [25, 63, 83], a popular class of boosting the encoder-decoder architecture can be regarded as incorporating various contextual modules [8, 40, 41, 69, 79, 83] into decoders. Consequently, the segmentation performance improvements are conferred by enabling each pixel to perceive information from other pixels in the input image. For instance, ASPP [8] and PPM [83] exploited dilated convolutions or pyramid pooling layers to aggregate pixel representations at different spatial positions. OCRNet [79] proposed to augment pixel representations with object-level representations. MCIBI++ [39, 41] differentiated the beyond-image contextual information from the within-image contextual information and thus improved pixel representations from the perspective of the whole dataset. To summarize, the existing contextual modules can be grouped into two batches, *i.e.*, multi-scale-driven [8, 70, 83] and similarity-driven [69, 79, 89] context schemes. Despite these methods achieved astonishing results, both schemes suffer from inadequate or ineffective contextual information aggregation as pixel relations are modeled with masses of predetermined priors. Specific to similarity-driven methods, contextual modules are designed to make pixels aggregate semantically similar contextual representations, which is inadequate for building the co-occurrent visual patterns. Also, some rigorous empirical analysis [5, 75] demonstrates that the contexts of pixels modeled by similarity-driven operations are indistinguishable for different query positions within an input image. As for multi-scale-driven approaches, pixel contexts are assumed to be fixed with certain geometric structures, which may compel pixels to capture some uninformative feature representations. Besides, previous studies [19, 67, 88] indicate that this fixed context assumption prevents the generalization of one-scenario-trained segmentors to other scenarios with unknown geometric transformations.

Different from conventional pipelines of building contextual modules, this paper investigates a more effective paradigm to build relationships among pixels. The proposed intervention-driven paradigm utilizes deletion diagnostics [16] to guide building pixel relations and thus aggregate more meaningful contextual information. In practice, there are two main challenges in constructing such a paradigm: (1) it is time-consuming to perform deletion diagnostics on all pixels and (2) it is memory-consuming to maintain a pixel relation matrix as one image contains plenty of pixels. Motivated by the observation that pixel relationships are closely connected to object relationships since a pixel semantic label is determined by which object region the pixel belongs to, we propose to simplify pixel-level relations modeling as object-level relations modeling to address the above-mentioned challenges.

Figure 1 depicts the diagram of our proposed intervention-driven relation network (IDRNet). We first group pixel representations into semantic-level representations according to predicted pseudo labels. Then, a relation matrix is constructed to make the semantic-level representations interact with each other and thus enhance themselves, where the relation matrix is updated by our deletion diagnostics mechanism. Finally, original pixel representations are augmented with the interacted representations and used to produce pixel labels. Compared with simply performing pixel-level deletion diagnostics, the GPU memory and computational complexity of our method can be significantly reduced. For example, on Cityscapes [17], we only need to maintain a relation matrix with a size $19 \times 19$ rather than a relation matrix with an input image size of $1024 \times 512$ during training. As the proposed deletion diagnostics mechanism is designed for semantic-level representations, it is also worth noting that our paradigm is general and can be easily extended to broader applications in other vision tasks, *e.g.*, object detection [27] and instance segmentation [12].

Table 1 compares the segmentation performance of various context modeling paradigms on Cityscapes and ADE20K benchmark datasets under fair settings (*e.g.*, backbone networks and training schedules). As observed, a simple FCN segmentation framework equipped with our intervention-driven context module (termed *IDRNet*) can achieve $43.61\%$ and $79.91\%$ mIoU on the datasets, respectively, which outperforms both dominant similarity-driven and multi-scale-driven context aggregation schemes. To further demonstrate the superiority of our method, we also integrate the proposed context scheme into UPerNet segmentation framework (termed *IDRNet+*), and it is observed that our paradigm with ResNet-50 backbone yields more impressive results on both datasets under single-scale testing.

In principle, our paradigm is fundamentally different from and would complement most similarity-driven and multi-scale-driven contextual modules. The main contribution can be summarized as,

- To our best knowledge, for the first time, this paper presents an intervention-driven paradigm, *i.e.*, deletion diagnostics mechanism, to help model pixel relations.
- This paper indicates that, in semantic segmentation task, pixel-level relation modeling can be simplified as building semantic-level relations.
- Our intervention-driven context modeling paradigm is conscientiously designed. It can be seamlessly integrated into various state-of-the-art semantic segmentation frameworks and achieve consistent performance improvements on various benchmark datasets, including ADE20K, Cityscapes, COCO-Stuff, LIP and PASCAL-Context.

We expect that IDRNet can provide insights for the future development of semantic segmentation and other related vision tasks.

## 2 Related Work

### 2.1 Semantic Segmentation

Semantic segmentation is a fundamental computer vision task, which is an indispensable component for many applications such as autopilot and medical diagnosis. Modern semantic image segmentation methods (in deep learning era) are believed to be derived from FCN [51] which formulates semantic segmentation as an encoder-decoder architecture. After that, numerous efforts have been devoted to improving the pipeline. For example, some tailored encoders like Res2Net [26] and HRNet [66] are designed for semantic segmentation to replace classification networks [33, 72]. In addition, automating network engineering [10, 47, 49] is also one of the most efficient ways to boost the encoder structure. Since transformer [64] attains growing attention, many recent works [1, 15, 21, 30, 50, 55, 62, 67, 71, 78, 86] also introduce transformer structure to help the encoder extract more effective pixel representations. Furthermore, there are also plenty of studies focusing on utilizing co-occurrent patterns to boost the decoder performance, *e.g.*, modeling long-range dependency [5, 31, 37, 40, 69, 75, 76] and enlarging the receptive field [7, 8, 9, 74, 77, 83]. Besides, some researchers also find that introducing context-aware optimization objectives [2, 42, 84] can strengthen context cues. Apart from the above, some studies also choose to break new ground, *e.g.*, introducing contrastive learning mechanism [68, 85], formulating the instance, panoptic and semantic segmentation tasks in a unified framework [12, 13, 82], leveraging boundary information [3, 20, 57, 65, 81], to name a few.

This paper studies how to leverage co-occurrent patterns. The key differences are that (1) we transform the pixel relation modeling problem to the semantic-level relation modeling problem and (2) for the first time, we propose to introduce deletion diagnostics mechanism to help model pixel relations.

### 2.2 Pixel Relation Modeling

As there exist co-occurrent patterns (*i.e.*, how likely two semantic classes can exist in a same image) in semantic segmentation, pixel relation modeling becomes a crucial component of recent state-of-the-art semantic segmentation algorithms [5, 36, 40, 70, 69, 75, 77, 79, 83]. For example, non-local network [69] built non-local blocks for capturing long-range dependencies. GCNet [5] unified the non-local and squeeze-excitation operations into a general framework for aggregating more efficacious global contextual information. DNL [75] presented the disentangled non-local block to help learn more discriminative pixel representations. ISA [36] was motivated to improve the efficiency of the self-attention operations and proposed to factorize one dense affinity matrix as the product of two sparse affinity matrices. Besides, OCRNet [79] proposed to emphasize utilizing the contextual information in the corresponding object class. ISNet [40] further improved OCRNet by incorporating image-level and semantic-level contextual information.

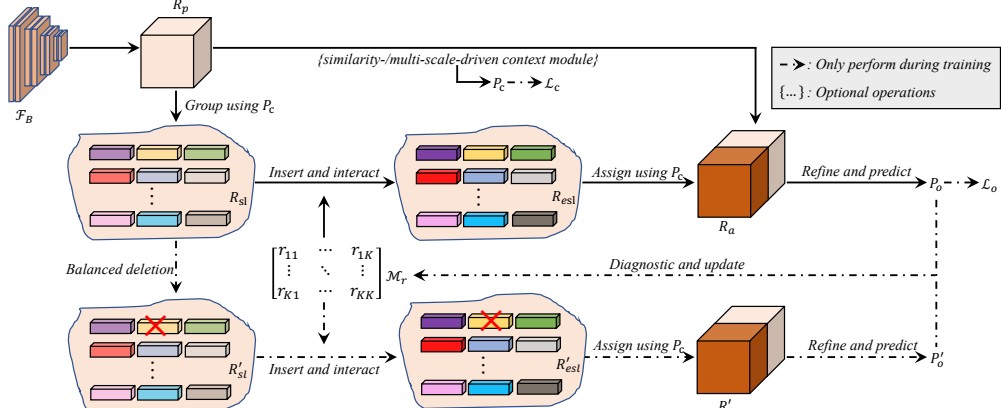

Figure 2: Illustration of our intervention-driven relation network (IDRNet). We first extract pixel representations $R_p$ using a backbone network $\mathcal{F}_B$, *e.g.*, ResNet [33] or SwinTransformer [50]. Then, $R_p$ is grouped into semantic-level representations $R_{sl}$ based on a coarse pixel prediction $P_c$. Next, we leverage a relation matrix $\mathcal{M}_r$ to make $R_{sl}$ interact with each other and further obtain enhanced semantic-level representations $R_{esl}$. Finally, $R_p$ is augmented by $R_{esl}$ and utilized for final pixel predictions $P_o$. After each training iteration, the deletion diagnostics is conducted to update $\mathcal{M}_r$.

Succinctly, these contextual modules utilize feature similarities or fixed geometric structures to model pixel relations, which injects an unavoidable human priors bias. Distinctively, this paper introduces deletion diagnostics mechanism to guide pixel relation modeling.

## 2.3 Object Relation Modeling

In object detection, co-occurrence is first applied in DPM [23] to refine object scores. After that, several works [14, 24, 53] were proposed to take object position and size into account for better building object relations in pre-deep learning era. In deep learning era, object relation modeling is also one of the most popular techniques to augment object representations. For instance, relation networks [34] extended original attention module to model both object similarities and relative geometry as the relations between objects. Motivated by relation networks, SGRN [73] took advantage of graph convolutional networks [43] to better bridge object regions by defining them as graph nodes. SMN [11] proposed to adopt a memory mechanism to perform object-object context reasoning.

Considering that each semantic-level representation in our paradigm plays a similar role to an object region in object detection, our intervention-driven relation modeling paradigm can also be extended to object detection tasks to help object relation modeling.

## 3 Methodology

### 3.1 Overview of IDRNet

Figure 2 illustrates our proposed intervention-driven relation network (IDRNet). The initial pixel-wise representations are grouped into semantic-level representations to facilitate relationship modeling. In detail, the semantic-level representations are first processed with a feature enhancement module and a feature interaction module with the guidance of a relation matrix $\mathcal{M}_r$, where the relation matrix is learned with the proposed deletion diagnostics paradigm. We subsequently augment the initial pixel representations with the processed semantic-level representations for final predictions.

**Semantic-level Representation Generation.** Following conventional FCN-based architecture [51], we first adopt a backbone network $\mathcal{F}_B$ to extract the pixel representations $R_p \in \mathbb{R}^{Z \times \frac{H}{8} \times \frac{W}{8}}$ from an input image $\mathcal{I} \in \mathbb{R}^{3 \times H \times W}$, where $H$ and $W$ are height and width of the image and $Z$ is the number of feature channels. To facilitate the seamless integration of our paradigm with existing segmentation frameworks, we formulate the pseudo-label generation process as follows,

$$P_c = \mathcal{F}_C(\mathcal{C}_e(R_p)), \ Y_c = \operatorname{argmax}(P_c), \tag{1}$$

where $\mathcal{C}_e$ denotes an identity function (instantiated as FCN [51]) or a previous similarity- / multi-scale-driven context module (instantiated as OCRNet [79] and PSPNet [83]) and $\mathcal{F}_C$ is a convolution layer

to project pixel representations into probability space. $P_c \in \mathbb{R}^{K \times \frac{H}{8} \times \frac{W}{8}}$ is the predicted probabilities of $K$ classes and $Y_c \in \mathbb{R}^{\frac{H}{8} \times \frac{W}{8}}$ is the pseudo label.

As it is computationally impracticable to direct model pixel-level relationships with deletion diagnostics, we propose to simplify pixel-level relations modeling as semantic-level relations modeling, *i.e.*, we group $R_p$ into semantic-level representations $R_{sl}$ by aggregating the corresponding input pixel representations with the same pseudo label,

$$R_{sl} = \left\{ \mathcal{F}_M \left( \{ R_{p,[*,\, i,\, j]} \mid Y_{c,[i,\, j]} = k \} \right), k \in [0, K) \right\}, \tag{2}$$

where $R_{sl} \in \mathbb{R}^{N_e \times Z}$ and $N_e$ is the number of classes existed in $\mathcal{I}$. $\mathcal{F}_M$ is a weighted summation operation to merge pixel representations with the corresponding probabilities in $P_c$ as the weights.

**Feature Enhancement.** We further introduce a feature enhancement module to improve the discrimination of $R_{sl}$ to help augment pixel representations $R_p$. Specifically, we modify Eq (2) as

$$R_{sl} = \left\{ \mathcal{F}_M \left( \{ R_{p,[*,\, i,\, j]} \oplus R_{ie,[k,*]} \mid Y_{c,[i,\, j]} = k \} \right), k \in [0, K) \right\}, \tag{3}$$

where $\oplus$ denotes a concatenation operation and $R_{ie} \in \mathbb{R}^{K \times Z}$ denotes discriminative vectors for each class in the training dataset. In our experiments, we investigate two strategies to obtain $R_{ie}$. The first one is to yield a random orthogonal matrix [52] to represent $R_{ie}$, while the second one is to learn a dataset-level representation for each category to construct $R_{ie}$ [39]. For the second strategy, we simplify $R_{ie}$ update strategy for class id $k$ as

$$R_{ie,[k,*]} = R_{ie,[k,*]} \times (1 - m) + R_{gt} \times m, \tag{4}$$

where $m$ is an update momentum and set as $0.1$ by default. Before training, we simply initialize $R_{ie}$ as a zero matrix and $R_{gt} \in \mathbb{R}^{1 \times Z}$ is calculated as

$$R_{gt} = \left\{ \text{Mean}(\{ R_{p,[*,\, i,\, j]} \mid GT_{[i,\, j]} = k \}), k \in [0, K) \right\}, \tag{5}$$

where $GT$ denotes the ground truth segmentation mask and $\text{Mean}(\cdot)$ is used to average the extracted pixel representations with the same ground truth label.

**Feature Interaction.** After the feature enhancement module, we propose to make $R_{sl}$ interact with each other to enrich themselves by using a semantic-level relation matrix $\mathcal{M}_r \in \mathbb{R}^{K \times K}$, which is updated by the proposed deletion diagnostics mechanism. The procedure of the feature interaction module can be represented as

$$R_{esl} = \text{Softmax}(\mathcal{T}(\mathcal{M}_r), dim = 1) \otimes R_{sl}, \tag{6}$$

where we adopt $\otimes$ to denote matrix multiplication and $R_{esl} \in \mathbb{R}^{N_e \times Z}$ is the enhanced semantic-level representations. $\mathcal{T}$ is employed to transform $\mathcal{M}_r$ to adapt the shape of $R_{sl}$, and it is formulated as

$$\hat{\mathcal{M}}_r = \mathcal{T}(\mathcal{M}_r) = \{ \mathcal{S}(\mathcal{M}_{r,[i,j]}, t) \mid i, j \in \{class\ ids\ in\ R_{sl}\} \}, \tag{7}$$

where $\mathcal{S}$ is introduced to control that there are no interactions between two semantic-level representations whose relationship in $\mathcal{M}_r$ is weak. $t = 0$ is a threshold utilized to identify those weak relationships. In practice, we set $\mathcal{M}_{r,[i,j]}$ be negative infinity in the calculation if $\mathcal{M}_{r,[i,j]}$ is smaller than $t$. Subsequently, we re-arrange $R_{esl}$ into the original pixel representation shape to augment $R_p$,

$$R_{a,[*,i,j]} = R_{esl,[k^{idx},*]} \ if \ Y_{c,[i,j]} = k, \ k \in \{class\ ids\ in\ R_{sl}\}, \tag{8}$$

where $k^{idx}$ is the corresponding matrix index for class id $k$ in $\mathcal{R}_{sl}$. $R_a \in \mathbb{R}^{Z \times \frac{H}{8} \times \frac{W}{8}}$ are the feature representations initialized as a zero matrix, and we fill it with the representations in $R_{esl}$ according to the pseudo labels in each pixel position.

**Final Prediction.** We use $R_a \in \mathbb{R}^{Z \times \frac{H}{8} \times \frac{W}{8}}$ to augment $R_p$ for final predictions,

$$\widetilde{R} = R_a \oplus \mathcal{C}_e(R_p), R = SA(\widetilde{R}), \tag{9}$$

where we use the self-attention mechanism $SA$ [64] to further process the augmented representations to balance the diversity and discriminative of the pixels with the same class. Based on $R$, the final pixel predictions $P_o \in \mathbb{R}^{K \times H \times W}$ is produced as

$$P_o = \mathcal{U}_{8 \times}(\mathcal{F}_O(R)), \tag{10}$$

where $\mathcal{F}_O$ is implemented by a convolution layer, which is utilized to produce pixel class probabilities and $\mathcal{U}$ is a bilinear interpolation operation.

During training, the overall objective function for backpropagation algorithm [45, 58] is defined as

$$\mathcal{L} = \alpha\mathcal{L}_c + \mathcal{L}_o, \tag{11}$$

where $\mathcal{L}_c$ and $\mathcal{L}_o$ are the cross entropy losses between $P_c$ and $GT$, and $P_o$ and $GT$, respectively. $\alpha$ is a hyperparameter for balancing $\mathcal{L}_c$ and $\mathcal{L}_o$. We empirically set it as $0.4$ in our experiments.

### 3.2 Deletion Diagnostics

Deletion diagnostics is proposed to update the relation matrix $\mathcal{M}_r$, which is initialized as an identity matrix. As shown in Figure 2, we randomly delete one semantic-level representation in $R_{sl}$ with a deletion distribution $Prob$ and obtain $R'_{sl}$. The deletion probability $Prob_i$ of the $i$-th semantic class representation is related to the number of times (*i.e.*, $count_i$) that this category has been deleted and can be represented as (termed balanced deletion in Figure 2)

$$Prob_i = \frac{\frac{1}{count_i}}{\sum \frac{1}{count_i}}, i \in \{class\ ids\ in\ R_{sl}\}. \tag{12}$$

We calculate the pixel predictions $P'_o$ by re-conducting Eq (6) - (10) with $R'_{sl}$ as input and further calculate the pixel-wise cross-entropy loss $l$ and $l'$ for $P_o$ and $P'_o$, respectively. After that, we extract the loss values from $l$ with respect to semantic class id $j$ as follows,

$$l_j = \{l_{[h,w]}|GT_{[h,w]} = j\}. \tag{13}$$

We also conduct the same operation on $l'$ to obtain $l'_j$.

Given the extracted two sets of loss values $l_j$ and $l'_j$, we can model the relationship between the deleted semantic class id $i$-th and one reserved class id $j$ in $R_{sl}$. Specifically, we calculate the mean and variance differences between $l_j$ and $l'_j$,

$$\begin{aligned} r^{i,j}_{mean} &= \frac{\sum l'_j - \sum l_j}{K \times H \times W}, \\ r^{i,j}_{var} &= \frac{\sum(l'_{j,[\cdot]} - \bar{l}'_j)^2 - \sum(l_{j,[\cdot]} - \bar{l}_j)^2}{K \times H \times W}, \end{aligned} \tag{14}$$

where $\bar{l}_j$ and $\bar{l}'_j$ denote the mean of $l_j$ and $l'_j$, respectively. Eq (14) indicates that the built semantic-level relations are supposed to help pixel recognition from the perspective of the mean and variance of the loss variation. To utilize such modeled relationship, we maintain two relation matrices $\mathcal{M}_{r_{mean}}$ and $\mathcal{M}_{r_{var}}$ during training and update them as

$$\begin{aligned} \mathcal{M}_{r_{mean},[i,j]} &= m_{mean} \times r^{i,j}_{mean} + (1 - m_{mean}) \times \mathcal{M}_{r_{mean},[i,j]}, \\ \mathcal{M}_{r_{var},[i,j]} &= m_{var} \times r^{i,j}_{var} + (1 - m_{var}) \times \mathcal{M}_{r_{var},[i,j]}, \end{aligned} \tag{15}$$

where $m_{mean}$ and $m_{var}$ are the momentum for two matrices and they are both empirically set as $0.1$. By re-conducting Eq (6)-(8) with $\mathcal{M}_{r_{mean}}$ and $\mathcal{M}_{r_{var}}$ as the relation matrix, we can obtain $R_{a,mean}$ and $R_{a,var}$, respectively. Finally, we combine them to obtain $R_a$ in Eq (9): $R_a = R_{a,mean} \oplus R_{a,var}$.

### 3.3 Discussion of IDRNet

The procedure of deletion diagnostics looks a lot like Dropout [61], while they are totally different mechanisms. Dropout is a mechanism to prevent deep neural networks from overfitting, while our approach introduces deletion diagnostics to help model pixel relations. In addition, our work also relates closely to some recent segmentation approaches, which extract semantic-level representations to augment pixel representations, *e.g.*, OCRNet [79] and ISNet [40]. Whereas, these methods only use feature similarities to make connections between pixels, which is distinct from our paradigm as it is intervention-driven and without masses of predetermined priors.

Our paradigm simplifies pixel relations modeling as object relations modeling. Benefiting from this simplification, deletion diagnostics enables the network to directly utilize its objective function to examine whether one class $i$ can help recognize another class $j$. The key idea behind this mechanism is that loss value of class $j$ should increase after deleting class $i$ if class $i$ can facilitate the prediction

Table 2: Performance improvements on various benchmark datasets with different segmentation frameworks after leveraging our intervention-driven paradigm to model pixel relations.

| Method | Backbone | Stride | ADE20K (*train / val*) | COCO-Stuff (*train / test*) | Cityscapes (*train / val*) | LIP (*train / val*) |
|---|---|---|---|---|---|---|
| FCN [51] | ResNet-50 | 8× | 36.96 | 31.76 | 75.16 | 48.63 |
| FCN+IDRNet (*ours*) | ResNet-50 | 8× | 43.61 (**+6.65**) | 38.64 (**+6.88**) | 79.91 (**+4.75**) | 51.24 (**+2.61**) |
| PSPNet [83] | ResNet-50 | 8× | 42.64 | 37.40 | 79.05 | 51.94 |
| PSPNet+IDRNet (*ours*) | ResNet-50 | 8× | 44.02 (**+1.38**) | 39.13 (**+1.73**) | 79.88 (**+0.83**) | 53.29 (**+1.35**) |
| DeeplabV3 [8] | ResNet-50 | 8× | 43.19 | 37.63 | 79.62 | 52.35 |
| DeeplabV3+IDRNet (*ours*) | ResNet-50 | 8× | 44.75 (**+1.56**) | 39.31 (**+1.68**) | 80.69 (**+1.07**) | 53.87 (**+1.52**) |
| UPerNet [70] | ResNet-50 | 8× | 43.02 | 37.65 | 79.08 | 52.95 |
| UPerNet+IDRNet (*ours*) | ResNet-50 | 8× | 44.84 (**+1.82**) | 39.35 (**+1.70**) | 80.81 (**+1.73**) | 54.00 (**+1.05**) |

Table 3: Ablation study on the design of IDRNet. IE-Orthogonal: feature enhancement with orthogonal matrix. IE-DL: feature enhancement with dataset-level representations. BD: balanced deletion in deletion diagnostics. RD: random deletion in deletion diagnostics.

| IE-Orthogonal | IE-DL | $\mathcal{M}_{r_{mean}}$ | $\mathcal{M}_{r_{var}}$ | BD | RD | ADE20K (*train / val*) | PASCAL-Context (*train / val*) | COCO-Stuff (*train / test*) |
|---|---|---|---|---|---|---|---|---|
| ✗ | ✗ | ✗ | ✗ | ✗ | ✗ | 36.96 | 45.48 | 31.76 |
| ✗ | ✗ | ✓ | ✓ | ✓ | ✗ | 42.82 | 52.65 | 37.79 |
| ✗ | ✓ | ✓ | ✓ | ✓ | ✗ | **43.61** | **52.97** | 38.64 |
| ✓ | ✗ | ✓ | ✓ | ✓ | ✗ | 42.99 | 52.80 | **38.86** |
| ✗ | ✓ | ✗ | ✓ | ✓ | ✗ | 42.74 | 52.86 | 38.06 |
| ✗ | ✓ | ✓ | ✗ | ✓ | ✗ | 43.13 | 52.66 | 37.89 |
| ✗ | ✓ | ✓ | ✓ | ✗ | ✓ | 42.90 | 52.92 | 37.59 |

of class $j$, and the increased value can be adopted to measure the importance of class $i$ to $j$. Therefore, there is no potential predetermined priors bias in the process of deletion diagnostics. As a comparison, the geometric transformations of multi-scale-driven context schemes are manually set, which tends to aggregate some ineffective information and lacks of generalization. Similarity-driven context schemes are assumed to aggregate semantically similar pixel representations, which results in ignoring other dissimilar but effective pixel representations for building co-occurrent patterns, *e.g.*, sky pixels for airplane pixels. In a nutshell, our intervention-driven paradigm is more effective compared to the existing similarity- / multi-scale-driven context modules.

## 4 Experiments

### 4.1 Experimental Setup

**Benchmark Datasets.** Our approach is validated on five popular semantic segmentation benchmark datasets, including ADE20K [87], COCO-Stuff [4], Cityscapes [17], LIP [29] and PASCAL-Context [22]. In detail, ADE20K is one of the most well-known datasets for scene parsing, which contains 150 stuff/object category labels. There are 20K/2K/3K images for training, validation and test set, respectively in the dataset. COCO-Stuff is a challenging dataset which provides rich annotations for 91 thing classes and 91 stuff classes. It consists of 9K/1K images in the training and test sets. Cityscapes has 5K high-resolution annotated urban scene images, with 2,975/500/1,524 images for training/validation/testing. It covers 19 challenging categories, such as traffic sign, vegetation and rider. LIP mainly focuses on single human parsing and contains 50K images with 19 semantic human part classes and 1 background class. Its training, validation and test sets separately involve 30K/10K/10K images. PASCAL-Context involves 59 foreground classes and a background class for scene parsing. The dataset is divided into 4,998 and 5,105 images for training and validation.

**Implementation Details.** Our algorithm is implemented in PyTorch [54] and SSSegmentation [38]. For backbone networks (*i.e.*, encoders), they are all pretrained on ImageNet dataset [59]. Remaining layers (*i.e.*, decoders) are initialized by the default initialization strategy in PyTorch. As for the data augmentation, some common data augmentation techniques are utilized, including random horizontal flipping, random cropping, color jitter and random scaling (from 0.5 to 2). The learning rate is scheduled by polynomial annealing policy with factor $(1 - \frac{iter}{total\_iter})^{0.9}$.

Specific to ADE20K, we set learning rate, crop size, batch size and training epochs as $0.01$, $512 \times 512$, 16 and 130, respectively. Specific to COCO-Stuff, learning rate, crop size, batch size and training epochs are set as $0.001$, $512 \times 512$, 16 and 110, respectively. As for LIP, we set learning rate, crop size, batch size and training epochs as $0.01$, $473 \times 473$, 32 and 150, respectively. As for Cityscapes, learning rate, crop size, batch size and training epochs are set as $0.01$, $512 \times 1024$, 8 and 220, respectively. Specific to PASCAL-Context, we set learning rate, crop size, batch size and training epochs as $0.004$, $480 \times 480$, 16 and 260, respectively.

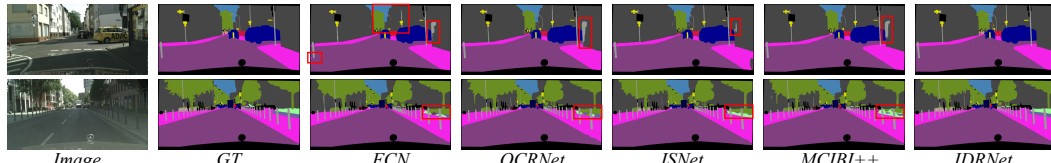

|  | Image | GT | FCN | OCRNet | ISNet | MCIBI++ | IDRNet |

Figure 3: Qualitative comparison with FCN, OCRNet, ISNet and MCIBI++ on Cityscapes dataset.

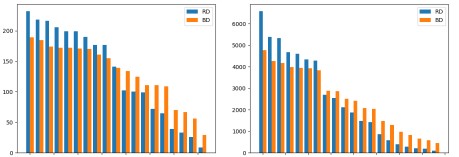

| Update strategy of $\mathcal{M}_r$ | Backbone | mIoU on ADE20K |
|---|---|---|
| FCN | ResNet-50 | 36.96 |
| FCN + BP-driven $\mathcal{M}_r$ | ResNet-50 | 40.35 |
| FCN + DD-driven $\mathcal{M}_r$ | ResNet-50 | 43.61 |

Figure 4: Compare category sampling frequency of balanced deletion (BD) and random deletion (RD) on Cityscapes (left) and LIP (right) benchmark dataset.

Table 4: Performance comparison between utilizing back propagation algorithm (*i.e.*, BP-driven $\mathcal{M}_r$) and using our deletion diagnostics (*i.e.*, DD-driven $\mathcal{M}_r$) to update the relation matrix $\mathcal{M}_r$.

**Evaluation Metric.** Following conventions [7, 51], mean intersection-over-union (mIoU) is utilized for evaluation.

### 4.2 Ablation Study

**Performance Improvements.** As seen in Table 2, it is observed that introducing deletion diagnostics to help pixel relationship building can bring consistent performance improvements on different segmentation approaches (*i.e.*, FCN, PSPNet, DeeplabV3 and UPerNet) and benchmark datasets (*i.e.*, ADE20K, COCO-Stuff, Cityscapes and LIP). By way of illustration, our method boosts FCN, PSPNet, DeeplabV3 and UPerNet by 2.61%, 1.35%, 1.52%, 1.05% mIoU on LIP dataset, respectively.

**Feature Enhancement.** Feature enhancement is designed to make pixel representations with different classes more discriminative to help model relationship between semantic-level representations $R_{sl}$. As indicated in Table 3, we can observe that yielding random orthogonal matrix (IE-Orthogonal) and generating dataset-level representations (IE-DL) for feature enhancement are both promising strategies to boost segmentation performance. Specifically, IE-Orthogonal brings 0.17%, 0.15% and 1.07% mIoU gains for ADE20K, PASCAL-Context and COCO-Stuff, respectively. IE-DL achieves 0.79%, 0.32% and 0.85% mIoU improvements for the three datasets, respectively.

**Relation Matrix.** We also validate the effectiveness of the proposed relation matrix in Table 3. It is observed that $\mathcal{M}_{r_{mean}}$ and $\mathcal{M}_{r_{var}}$ both contribute to the performance improvements. In detail, $\mathcal{M}_{r_{mean}}$ makes our segmentor outperform baseline models (first row) by 6.17%, 7.18% and 6.13% on ADE20K, PASCAL-Context and COCO-Stuff, respectively. $\mathcal{M}_{r_{var}}$ boosts baseline models by 5.78%, 7.38% and 6.30% on the three datasets, respectively. The combination of $\mathcal{M}_{r_{mean}}$ and $\mathcal{M}_{r_{var}}$ finally yields 43.61%, 52.97% and 38.64% mIoU on the three datasets, respectively.

**Deletion Diagnostics.** In Table 4, we compare our proposed deletion diagnostics procedure (DD-driven $\mathcal{M}_r$) with back-propagation updating strategy (BP-driven $\mathcal{M}_r$). We can observe that adopting DD-driven $\mathcal{M}_r$ outperforms using BP-driven $\mathcal{M}_r$ by 3.26% mIoU on ADE20k. The result suggests that our deletion diagnostics strategy is more effective than the back-propagation updating strategy.

**Balanced Deletion.** Balanced deletion is utilized to help improve the being-diagnosed probabilities of uncommon categories in training dataset and thereby, deletion diagnostics can be performed evenly. In Table 3, we can observe that introducing the balanced deletion strategy can help improve the segmentation performance by 0.71%, 0.05% and 1.05% for ADE20K, PASCAL-Context and COCO-Stuff, respectively. Also, we count and compare the sampling frequency of each category when using balanced deletion (BD) and random deletion (RD) in Figure 4. We can observe that the sampling frequency of uncommon categories have increased after utilizing BD as we expected.

**Complexity Comparison.** We also present analysis on complexity of our intervention-driven context module following the settings of OCR [79]. In Table 5, it is observed that our IDRNet requires the least Parameters, FLOPS and GPU Memory, while achieving the best mIoU on the ADE20k dataset. Moreover, our IDRNet is complementary to other context modules. For example, when incorporated with PPM (termed "PPM+IDRNet" in the table), the Parameters, FLOPS, Time and GPU Memory only increase by 0.58 M, 39.78 G, 11.19 ms and 73.65 M, respectively, which shows that our proposed IDRNet is light and efficient during network inference. Note that few parameters

Table 5: Complexity comparison with existing context modules on a single RTX 3090 Ti GPU. The input feature map is with size $[1 \times 2048 \times 128 \times 128]$. Excepted for mIoU column, all numbers are the smaller the better.

| Context Module | Parameters | FLOPS | Time | GPU Memory | mIoU on ADE20K (%) |
|---|---|---|---|---|---|
| OCR [79] | 15.12M | 242.48G | 16.58ms | 617.24M | 42.47 |
| ASPP [8] | 42.21M | 674.47G | 41.98ms | 976.06M | 43.19 |
| PPM [83] | 23.07M | 309.45G | 21.45ms | 960.63M | 42.64 |
| UPerNet [70] | 34.75M | 500.76G | 36.51ms | 1429.18M | 43.02 |
| ANN [89] | 22.42M | 369.62G | 26.58ms | 1445.75M | 41.75 |
| CCNet [37] | 23.92M | 397.38G | 30.92ms | 986.28M | 42.48 |
| DNL [75] | 24.12M | 395.25G | 51.38ms | 2381.04M | 43.50 |
| IDRNet (*ours*) | 10.79M | 155.89G | 20.52ms | 365.66M | 43.61 |
| PPM+IDRNet (*ours*) | 23.65M | 349.23G | 32.64ms | 1034.28M | 44.02 |

Table 6: Compare segmentation performance in cross domain.

| Method | **Urban Scene Parsing** (*Cityscapes train*) | | **Human Parsing** (*LIP train*) |
|---|---|---|---|
| | *Dark Zurich val* | *Nighttime Driving test* | *CIHP val* |
| FCN [51] | 10.66 | 17.90 | 27.20 |
| FCN+IDRNet (*ours*) | 12.55 (**+1.89**) | 21.33 (**+3.43**) | 28.59 (**+1.39**) |
| DeeplabV3 [8] | 10.03 | 21.91 | 27.02 |
| DeeplabV3+IDRNet (*ours*) | 13.66 (**+3.63**) | 23.85 (**+1.94**) | 27.93 (**+0.91**) |
| UPerNet [83] | 11.06 | 17.67 | 26.84 |
| UPerNet+IDRNet (*ours*) | 11.42 (**+0.36**) | 21.13 (**+3.46**) | 27.46 (**+0.62**) |

increase in PPM+IDR as we adopt a shared $3 \times 3$ convolution layer to reduce the feature channels of the backbone outputs.

**Qualitative Results.** Figure 3 provides qualitative comparison of the proposed approach against FCN [51], OCRNet [79], ISNet [40] and MCIBI++[41] on representative examples in various benchmark datasets. It is observed that IDRNet can output better segmentation results.

### 4.3 Cross Domain Segmentation Performance.

Table 6 compares the cross-domain segmentation performance. It can be observed that IDRNet can also boost cross-domain segmentation performance, which demonstrates the generalization of our intervention-driven pixel relation modeling paradigm. For example, DeeplabV3+IDRNet trained on day-time domain (*i.e.*, Cityscapes) can also bring 3.63% and 1.94% mIoU gains to Dark Zurich [60] and Nighttime Driving [18] datasets which belong to night-time domain. In addition, after training FCN+IDRNet on single-human parsing scenarios (*i.e.*, LIP), it also boosts the segmentation performance on multi-human parsing scenarios (*i.e.*, CIHP [28]) by 1.39% mIoU.

### 4.4 Comparison with State-of-the-art Methods

Table 7 compares the quantitative results on four challenging benchmark datasets, including ADE20K *val*, LIP *val*, PASCAL-Context *val* and COCO-Stuff *test*. As for ADE20K *val*, it is observed that prior to this paper, Mask2Former [12] with Swin-Large backbone [41] achieves the state-of-the-art performance hitting 57.30% mIoU. By incorporating Mask2Former with IDRNet, it yields 58.22% mIoU which surpasses other state-of-the-art segmentation methods. Specific to LIP *val*, we can observe that among the previous SOTA methods, UPerNet+MCIBI++ with Swin-Large backbone obtains the state-of-the-art result, *i.e.*, 59.91% mIoU. Our method, UPerNet+IDRNet with Swin-Large backbone, outperforms it by 1.26% and reports the new SOTA result, 61.17% mIoU. With regard to PASCAL-Context *val*, our method, UPerNet+IDRNet with Swin-Large backbone, surpasses all the competitors, *i.e.*, 0.50% over UPerNet-MCIBI++, 5.51% over Segmenter, 8.68% over SETR, 9.71% over OCRNet, to name a few. On the subject of COCO-Stuff *test*, we can observe that our UPerNet+IDRNet with Swin-Large backbone reaches 50.50% mIoU which is 0.23% higher than previous state-of-the-art method UPerNet+MCIBI++.

### 4.5 Extend Deletion Diagnostics to Object Detection

In this section, we apply deletion diagnostics mechanism to the object detection frameworks to further demonstrate the effectiveness and generalization of our method.

**Implementation Details.** Our experiments are conducted on MS COCO dataset [48]. All algorithm implementations are based on PyTorch [54] and MMDetection [6], and we follow the default settings in MMDetection for model training and testing.

Table 7: State-of-the-art results on ADE20K, COCO-Stuff, LIP and PASCAL-Context dataset.

| Method | Backbone | Stride | ADE20K (*train / val*) | COCO-Stuff (*train / test*) | PASCAL-Context (*train / val*) | LIP (*train / val*) |
|---|---|---|---|---|---|---|
| PSPNet [83] | ResNet-101 | 8× | 43.29 | - | 47.80 | - |
| EMANet [46] | ResNet-101 | 8× | - | 39.90 | 53.10 | - |
| OCNet [80] | ResNet-101 | 8× | 45.45 | - | - | 54.72 |
| ACNet [35] | ResNet-101 | 8× | 45.90 | 40.10 | 54.10 | - |
| APCNet [31] | ResNet-101 | 8× | 45.38 | - | 54.70 | - |
| OCRNet [79] | ResNet-101 | 8× | 45.28 | 39.50 | 54.80 | 55.60 |
| ISNet [40] | ResNet-101 | 8× | 47.31 | 41.60 | - | 55.41 |
| UPerNet+MCIBI++ [41] | ResNet-101 | 8× | 47.93 | 41.84 | 56.82 | 56.32 |
| OCRNet [79] | HRNetV2-W48 | 4× | 45.66 | 40.50 | 56.20 | 56.65 |
| SETR [86] | ViT-Large | 16× | 50.28 | - | 55.83 | - |
| Segmenter [62] | ViT-Large | 16× | 53.63 | - | 59.00 | - |
| UPerNet [70] | BEiT-Large | 16× | 57.00 | - | - | - |
| UPerNet [70] | Swin-Large | 32× | 53.50 | - | - | - |
| UPerNet+MCIBI++ [41] | Swin-Large | 32× | 54.52 | 50.27 | 64.01 | 59.91 |
| MaskFormer [13] | Swin-Large | 32× | 55.60 | - | - | - |
| Mask2Former [12] | Swin-Large | 32× | 57.30 | - | - | - |
| K-Net+UPerNet [82] | Swin-Large | 32× | 54.30 | - | - | - |
| SegNeXt-L [30] | MSCAN-Large | 32× | 52.10 | 47.20 | 60.30 | - |
| Segformer [71] | MiT-B5 | 32× | 51.80 | 46.70 | - | - |
| UPerNet+IDRNet (*ours*) | Swin-Large | 32× | 54.67 | **50.50** | **64.51** | 61.17 |
| Mask2Former+IDRNet (*ours*) | Swin-Large | 32× | **58.22** | 49.96 | 61.65 | **61.86** |

Table 8: Performance improvements on two popular object detection frameworks and MS COCO benchmark dataset after leveraging deletion diagnostics to build object relations.

| Method | Backbone | Schedule | AP | $AP_{50}$ | $AP_{75}$ | $AP_s$ | $AP_m$ | $AP_l$ |
|---|---|---|---|---|---|---|---|---|
| Faster R-CNN [56] | ResNet-50-FPN | 1× | 37.4 | 58.1 | 40.4 | 21.2 | 41.0 | 48.1 |
| Faster R-CNN+IDRNet (*ours*) | ResNet-50-FPN | 1× | 38.2 (**+0.8**) | 59,7 | 41.3 | 22.0 | 42.0 | 49.2 |
| Mask R-CNN [32] | ResNet-50-FPN | 1× | 38.2 | 58.8 | 41.4 | 21.9 | 40.9 | 49.5 |
| Mask R-CNN+IDRNet (*ours*) | ResNet-50-FPN | 1× | 39.7 (**+1.5**) | 60.8 | 43.0 | 22.6 | 43.2 | 52.5 |

**Experimental Results.** As illustrated in Table 8, the proposed deletion diagnostics mechanism is integrated into two popular object detection frameworks, *i.e.*, Faster R-CNN [56] and Mask R-CNN [32]. We can observe that intervention-driven relation building for objects brings $0.8\%$ and $1.5\%$ AP improvements for the two detection frameworks, respectively.

## 5 Conclusion

This work studies how to utilize deletion diagnostics to model pixel relationships. The key contribution consists of two parts: (1) we simplify pixel relation modeling as object relation modeling and (2) we introduce deletion diagnostics to make the networks build object relationships by perceiving changes in the outputs of the objective functions. Compared to similarity-/multi-scale-driven context modeling paradigms, there are fewer human priors in our context modeling paradigm. Experimental results demonstrate the effectiveness of our proposed paradigm qualitatively and quantitatively.

## Acknowledgement

The work described in this paper was partially supported by grants from the National Key R&D Program of China (2022ZD0118101), the National Natural Science Fund (62201483) and the Research Grants Council of the Hong Kong Special Administrative Region, China (T45-401/22-N).

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
