# OpenReview forum: "IDRNet: Intervention-Driven Relation Network for Semantic Segmentation"
_NeurIPS.cc/2023/Conference — NeurIPS 2023 poster_

### Official Review · Reviewer_3dVM · 2023-07-03

**Soundness:** 3 good
**Presentation:** 3 good
**Contribution:** 3 good
**Rating:** 6
**Confidence:** 3

**Summary:**

This study focuses on the utilization of deletion diagnostics for modeling pixel relationships. The main contribution of this work can be divided into two parts. Firstly, the simplification of pixel relation modeling as object relation modeling. Secondly, the introduction of deletion diagnostics to enable networks to build object relationships by detecting changes in the outputs of objective functions. Experimental results qualitatively and quantitatively demonstrate the effectiveness of the proposed paradigm.

**Strengths:**

* The use of Deletion Diagnostics to update the relation matrix is a novel approach that considers the problem of category context in semantic segmentation from a different perspective than previous multi-scale driven and similarity-driven methods.

* Thorough ablation experiments were conducted to verify the impact of different designs of IDRNet on segmentation performance.

* The writing is clear and easily understandable for readers.

**Weaknesses:**

* The experiments of Mask2Former+IDRNet only report results on one dataset, ADE20K. As Mask2Former is a general segmentation framework, it is recommended that the authors also report the performance of the proposed model on the other three datasets for a comprehensive comparison.

* Additional experiments based on Mask2Former could be added to Table 2 to verify the effectiveness of IDRNet on a regular backbone.

**Questions:**

See Weaknesses.

**Limitations:**

There seems no discussion about limitations in the paper.

---

> ### Author Rebuttal · Authors · 2023-08-09
>
> We highly appreciate your insightful reviews and positive comments on our novel methodology, the extensive evaluation, and clear writting. Our responses are shown below.
>
>
> > **Q1: Missing results for Mask2Former+IDRNet on LIP/COCO-Stuff/Pascal Context.**
>
> Following your suggestion, we also conduct the experiments for Mask2Former+IDRNet on LIP, COCO-Stuff and Pascal Context datasets and the results are shown in the below table. We will add these experiments to Table 5 in the revised version.
>
> | Method             |   Backbone   | ADE20k  | LIP    | COCO-Stuff  | Pascal Context |
> | :---:              |   :---:      | :---:   | :---:  | :---:       | :---:          |
> | Mask2Former        |   Swin-Large | 57.30   | 60.37  | 48.08       | 60.67          |
> | Mask2Former+IDR |   Swin-Large | 58.22   | 61.86  | 49.96       | 61.65          |
>
> It is observed that Mask2Former+IDRNet also outperforms Mask2Former for 1.49%, 1.88% and 0.98% mIoU on LIP, COCO-Stuff and Pascal Context datasets, respectively, which further show the effectiveness of our method.
>
>
> > **Q2: Additional experiments based on Mask2Former with a regular backbone could be added to Table2.**
>
> Thanks for your suggestions. Additional experiments based on Mask2Former with a regular ResNet-50 backbone are conducted and reported in the below table. We will also add these experiments to Table 2 in the revised version.
>
> | Method             |   Backbone   | ADE20k  | LIP    | COCO-Stuff  | Cityscapes     |
> | :---:              |   :---:      | :---:   | :---:  | :---:       | :---:          |
> | Mask2Former        |   ResNet-50  | 47.20   | 51.23  | 37.22       | 79.40          |
> | Mask2Former+IDR |   ResNet-50  | 48.49   | 53.22  | 38.45       | 80.79          |
>
> It is observed that our Mask2Former+IDRNet consistently outperforms Mask2Former by more than 1% mIoU on various benchmark datasets with a regular ResNet-50 backbone.

---

### Official Review · Reviewer_dJpB · 2023-07-05

**Soundness:** 3 good
**Presentation:** 3 good
**Contribution:** 1 poor
**Rating:** 5
**Confidence:** 4

**Summary:**

This work presents IDRNet, a context aggregation scheme for semantic segmentation networks which is driven by intervention in the object query stage. The work is a follow up on OCR (object-contextual representations), but adds to the previous work by an intervention-based strategy to update, augment, or delete tokens. The updates to these tokens are based on semantic relations.

**Strengths:**

Presentation:
The paper is clear in its presentation of the concept, and the result. Tables and figures are self-explanatory and it is easy to follow the though process of the authors. The reviewer appreciates the effort spent in explaining the concept of this work.

Results:
As a scheme which is built to add to pixel representations, the results are quite impressive on five different datasets. The authors have also provided ablation studies detailing the effect of each separate block in the proposed algorithm.

Visulalizations:
The reviewer also appreciates the detail the authors have provided on viewing semantic relationships, which was the major motivation of their approach. Hence, it is safe to say that the motivation of the work and results go hand in hand.

**Weaknesses:**

Originality:
The concept of updating (deleting, augmenting, enhancing, "diagnosing") embeddings based on object relations might seem novel when you think of them in a convolution-based setting. However, I am afraid this concept is not original in transformer networks. Works such as Mask2Former do present queries which describe objects and the network learns (based on bipartite matching) to associate queries with the concept of object. There are also several other works [1], [2], [3] which prune, refine or update these tokens based on semantic relationships.

Complexity:
As the method adds additional compute to the network, the authors mention how much compute is added and also, more importantly, is Baseline+IDRNet better than simply scaling up the baseline to match this compute.

[1] Revisiting Token Pruning for Object Detection and Instance Segmentation
[2] Sparse Tokens for Dense Prediction - The Medical Image Segmentation Case
[3] DynaMITe: Dynamic Query Bootstrapping for Multi-object Interactive
Segmentation Transformer

**Questions:**

1. Please could describe how this method is different from other works performing token pruning
2. Please could you provide some analysis on complexity/runtime related to other context aggregation schemes such as OCR
3. Please could you provide some limitations of the approach

**Limitations:**

The authors have not addressed limitations of their method.

---

> ### Author Rebuttal · Authors · 2023-08-09
>
> We sincerely thank you for the comments. We hope that our responses could thoroughly address your concerns.
>
> > **Q1: How this method is different from other works performing token pruning.**
>
> We are glad to clarify the differences between our work and those token pruning works.
>
> - The objectives are different. The mentioned works aim to identify which are the important representations / tokens for network pruning, while our method aims to augment the image representations via utilizing and learning the relationships among different semantic representations.
>
> - The definition of representations / tokens are different. The tokens in [1] (released in June 2023) and [2] are flattened 2D image patches, which may ***contain the information of multiple, singe, or no objects***. [3] is an interactive segmentation algorithm and the queries are dynamically generated using image features and user clicks. The object queries in [4] are a set of learnable embeddings. As a comparison, the semantic-level representations in IDRNet are yielded by gathering the pixel representations with the same label and each semantic-level representation ***only contains the information of the corresponding category***.
>
> - The principles to build the relationships between representations / tokens are different.  [1]-[4] leverage the transformer and back-propagation to learn the ***statistical correlation*** between tokens / queries, while our IDRNet introduces the ***intervention-based*** deletion diagnostics procedure to update the relation matrix for the semantic-level representations. We make attempts to only adopt back-propagation to update the relation matrix in IDRNet, while the the mIoU on ADE20k drops from 43.61% to 40.35%, verifying the effectiveness of our intervention-based method.
>
> - The approach to learn the relationship are different. [1] utilizes a gating module (a 2-layer perceptron + Gumbel Softmax) to facilitate acceleration of transformer encoder and the learnable parameters in the gating module are supervised by a dynamic pruning ratio loss. Differently, the deletion diagnostics in IDRNet is to perform pixel relationship modeling and the relation matrix is updated by the changes of the objective function value, *e.g.*, the cross entropy loss.
>
> - Our intervention-driven strategy is orthogonal to the segmentation backbone design and ***can also be intergrated into transformer-based backbones***. Particularly, we integrate IDR with Mask2Former [4]. We can observe that IDR brings 0.92%, 1.49%, 1.88% and 0.98% mIoU improvements on different benchmark datasets, suggesting that IDR is complementary to Mask2Former.
>
> | Method             |   Backbone   | ADE20k  | LIP    | COCO-Stuff  | Pascal Context |
> | :---:              |   :---:      | :---:   | :---:  | :---:       | :---:          |
> | Mask2Former        |   Swin-Large | 57.30   | 60.37  | 48.08       | 60.67          |
> | Mask2Former+IDR |   Swin-Large | 58.22   | 61.86  | 49.96       | 61.65          |
>
> In summary, our method is ***inherently different*** from [1]-[4].
>
> [1] Liu et al. Revisiting Token Pruning for Object Detection and Instance Segmentation. arXiv 2023.
> [2] Zhou et al. Sparse Tokens for Dense Prediction-The Medical Image Segmentation Case. 2022.
> [3] Rana et al. DynaMITe: Dynamic Query Bootstrapping for Multi-object Interactive Segmentation Transformer. arXiv 2023.
> [4] Cheng et al. Masked-attention mask transformer for universal image segmentation. CVPR 2022.
>
> > **Q2: Provide some analysis on complexity/runtime related to other context aggregation schemes such as OCR.**
>
> We present analysis on complexity/runtime of our intervention-driven context module following the settings of OCR. Specifically, we compare the Params, FLOPS, Time and Memory of our IDRNet with previous context modules in the below table.
>
> | Context Module | Params | FLOPS |  Time    | Memory  |  mIoU on ADE20k (%)  |
> | :---:             |  :---:   |  :---:    |  :---:   | :---:    |  :---: |
> | OCR       |  15.12M  |  242.48G  |  16.58ms | 617.24M  |  42.47 |
> | ASPP      |  42.21M  |  674.47G  |  41.98ms | 976.06M  |  43.19 |
> | PPM       |  23.07M  |  309.45G  |  21.45ms | 960.63M  |  42.64 |
> | UperNet   |  34.75M  |  500.76G  |  36.51ms | 1429.18M |  43.02 |
> | ANN       |  22.42M  |  369.62G  |  26.58ms | 1445.75M |  41.75 |
> | CCNet     |  23.92M  |  397.38G  |  30.92ms | 986.28M  |  42.48 |
> | DNL       |  24.12M  |  395.25G  |  51.38ms | 2381.04M |  43.50 |
> | IDR (*ours*)      |  10.79M  |  155.89G  |  20.52ms | 365.66M  |  43.61 |
> | PPM+IDR (*ours*)  |  23.65M  |  349.23G  |  32.64ms | 1034.28M |  44.02 |
>
> It is observed that IDR module requires the least Params, FLOPS and GPU Memory,  while achieving the best mIoU on the ADE20k dataset. Moreover, our IDR module is complementary to other context modules. For example, when incorporated with PPM (***PPM+IDR***), the Params, FLOPS, Time and Memory only increase by 0.58 M, 39.78 G, 11.19 ms and 73.65 M, respectively, which shows that our proposed IDR is light and efficient during network inference. Note that few parameters increase in ***PPM+IDR*** as we adopt a shared $3 \times 3$ convolution to reduce the feature channels of the backbone outputs.
>
> > **Q3: Provide some limitations of the approach.**
>
> We briefly discuss some limitation and future works of our approach.
>
> - Some modules in IDRNet needs to perform forward function twice to update $\mathcal{M}$ during training, Thus, IDRNet would require more training cost compared to the previous context modules.
> - Due to the limited resource, we did not explore the effectiveness of IDRNet on the large-scale vision foundation models like InternImage $^{[1]}$.
> - In the future, we will investigate the effectiveness of our IDRNet on the domain-specific problem, like medical image segmentation.
>
> [1] Wang et al. Internimage: Exploring large-scale vision foundation models with deformable convolutions, CVPR 2023.

---

> > ### Comment · Reviewer_dJpB · 2023-08-17
> >
> > Thanks for providing the clarifications. Please could you let me know how you apply IDRNet to Mask2Former? In this case, did you apply the diagnostics to the pixel representations? Did you use mask embeddings directly as your object embeddings?
> >
> > Furthermore, could you please clarify in the OCR analysis table what the backbone is for each experiment? Is it the same backbone for OCR and IDR?

---

> > > ### Author Response · Authors · 2023-08-19
> > >
> > > > **Please could you let me know how you apply IDRNet to Mask2Former? In this case, did you apply the diagnostics to the pixel representations? Did you use mask embeddings directly as your object embeddings?**
> > >
> > > We are glad to explain how we apply IDRNet to Mask2Former.
> > >
> > > Actually, **we did not make extra designs for applying IDRNet to Mask2Former**. Specifically, we simply add IDR module after Swin-Large backbone to enhance the backbone output features and then, input the enhanced features to Mask2FormerHead to obtain mask predictions and the corresponding classification scores.
> > >
> > > Thus, when applying IDRNet to Mask2Former, we still **apply the deletion diagnostics to the semantic-level representations** to update the relation matrix $\mathcal{M}$ as described in Section 3 in the submitted manuscript. And we did not use the mask embeddings directly as our object embeddings since **we leverage the pixel-level pseudo labels to calculate the semantic-level representations**.
> > >
> > > To provide full details of our method, our code will be made publicly available.
> > >
> > > > **Could you please clarify in the OCR analysis table what the backbone is for each experiment? Is it the same backbone for OCR and IDR?**
> > >
> > > We are glad to clarify this point. In the OCR analysis table, all listed context modules, including OCR (we refer to mmsegmentation to implement OCR with ResNet backbone) and IDR, **use the same backbone, _i.e._, ResNet-50**.

---

> > > > ### Comment · Reviewer_dJpB · 2023-08-19
> > > >
> > > > Thanks for your clarifications. I still feel there is some novelty lacking, but overall I will upgrade my rating to borderline.

---

### Official Review · Reviewer_q3X9 · 2023-07-06

**Soundness:** 4 excellent
**Presentation:** 2 fair
**Contribution:** 4 excellent
**Rating:** 6
**Confidence:** 5

**Summary:**

This paper presents a context relation modeling method, which can be incorporated with existing semantic segmentation model. Instead of modeling pixel-level relation, it models class-level semantic relation by weighted averaging features at each location and deletion diagnostics. The proposed module is flexible and tested with different network backbone and segmentation architectures, which observes consistent improvements on several widely used datasets. Furthermore, the proposed context relation module can be applied in object detection, that stronger results on MS COCO are shown based on Faster-RCNN.

**Strengths:**

+ This paper presents a context relation module, which can be combined with previous semantic segmentation models. It is used to enhance the learned features and perform features interaction with global co-cocurrence. Besides, in order to overcome the heavy computation issue, this work proposes to use semantic relation to replace pixel relation.

+ This paper conduct extensive experimental results, including 4 popular datasets. Besides, this work is combined with many previous segmentation models. For example, even though PSPNet can aggregate context information, the semantic relation module can further improve its performance.

+ Comparison with previous state-of-the-art demonstrates the effectiveness of the proposed module.

+ This work is also compatible with object detection framework. It should be also feasible to apply the proposed module in semantic instance segmentation.

**Weaknesses:**

- Too many symbols in section 3. For some details, I cannot follow.
In Eq 3, what is the output dimension for R_{sl} ?
In Eq 6, why is R_{esl} a Ne xZ matrix, as M_r is KxK ?
In Eq 8, what are the values in R_a if Y_c != k ?
In Eq 9, what does the \bigoplus operation represent ?

- Figure 5 is hard to see the graph connection, for example, the rider class. For sky, it is connected to truck, building, road, which does not seem to make sense.

- In Table 5, why not show the performance for Mask2Former+IDRNet on LIP/COCO-Stuff/PASCAL-Context datasets?

- Suggest to list all the comparing methods.

**Questions:**

1. Actually, you do not have to use deletion diagnostics to update M.  What is the motivation of deletion diagnostics? What kind of issues does the proposed deletion aim to address?


**Limitations:**

See weakness section.

---

> ### Author Rebuttal · Authors · 2023-08-09
>
> We highly appreciate your insightful reviews and positive comments on our methodology design, extensive evaluation, and framework generalizability. Our responses are shown below.
>
>
> > **Q1: Explanation of symbols in Section 3.**
>
> We are glad to explain the meaning of these symbols and will add these explanations and also list all the comparing methods following your suggestation in the revised version.
>
> - The output dimension for $R_{sl}$ in Eq (3) is $\mathbb{R}^{N_e \times 2Z}$, where $N_e$ is the number of classes existed in the input image and $Z$ is the number of channels of $R_p$ (or $R_{ie}$). When introducing the ***Feature Enhancement*** module, we use a $1 \times 1$ convolution layer to make $R_{sl}$ be with the shape of $N_e \times Z$ after Eq (3) to avoid introducing too many extra parameters in IDRNet.
>
> - In Eq (6), we first leverage $\mathcal{T}$ (see Eq (7) for more details) to transform $\mathcal{M}_r \in \mathbb{R}^{K \times K} $ into $\mathcal{\hat{M}}_r \in \mathbb{R}^{N_e \times N_e}$ for matching the shape of  $R\_{sl}$ and thereby, $\hat{\mathcal{M}}\_{r}$ only contains the relation values of the classes existed in the current input image. As a result, the shape of $R\_{esl}$ is $N\_{e} \times Z$.
>
> - $R_a \in \mathbb{R}^{Z \times \frac{H}{8} \times \frac{W}{8}}$ is the feature representations by re-arranging $R_{esl}$. Specifically, we initialize $R_a$ as a zero matrix and fill it with the representations in $R_{esl}$ according to the pseudo labels in each pixel position. For example, if the labels in the input image $I$ are {$0, 1, 5, 7$}, $R_{esl}$ will be {$\{R_{esl}^{cls=0}, R_{esl}^{cls=1}, R_{esl}^{cls=5}, R_{esl}^{cls=7}\}$} in Eq (6). And if class id of $R_{a, [\*, 0, 0]}$ is 5, we will set $R_{a, [\*, 0, 0]} = R_{esl}^{cls=5}$ in Eq (8). Thus, $Y_{c, [i,j]}$ will always be equal to one of the class ids existed in $R_{esl}$ and it is impossible for the case of $Y_{c, [i,j]} \neq k$.
>
> - $\oplus$ indicates a concatenation operation as explained after Eq (3). In the revised version, we will further explain it after Eq (9) to easy understanding.
>
> > **Q2: Figure 5 is hard to see the graph connection.**
>
> In Figure 5, we show the relationships graph without adding the corresponding connection weights, which may cause some confusion. Actually, "sky is connected to truck, building and road" is reasonable in the Cityscape datasets since there are some images containing both "truck, building and road" and "sky". However, the connection weight of "truck" and "sky" in $\mathcal{M}$ is ***much smaller*** than the one of "truck" and "road". Therefore, the pixel representations of "truck" will aggregate much more information from the "road" rather than "sky". In the revised version, we will mark relation weights between the classes and modify the graph to show only the top k connections to better visualize the relationships between the classes.
>
> > **Q3: Missing results for Mask2Former+IDRNet on LIP/COCO-Stuff/Pascal Context.**
>
> In Table 5, as the performance of UPerNet+IDRNet has already achieved the state-of-the-art performance on LIP/COCO-Stuff/Pascal Context, we did not show the performance of Mask2Former+IDRNet on these datasets. Following your suggestion, we have conducted the experiments for Mask2Former+IDR on LIP, COCO-Stuff and Pascal Context datasets and the results are shown below, which will be added in the revised version.
>
> | Method             |   Backbone   | ADE20k  | LIP    | COCO-Stuff  | Pascal Context |
> | :---:              |   :---:      | :---:   | :---:  | :---:       | :---:          |
> | Mask2Former        |   Swin-Large | 57.30   | 60.37  | 48.08       | 60.67          |
> | Mask2Former+IDR |   Swin-Large | 58.22   | 61.86  | 49.96       | 61.65          |
>
> > **Q4: Motivation and benefit of deletion diagnostics.**
>
> Thanks for your comment. First, there are indeed different ways to update $\mathcal{M}$, such as back-propagation. In the below table, we compare our proposed deletion diagnostics procedure with back-propagation updating strategy. We can observe that deletion diagnostics strategy outperforms back-propagation by 3.26% mIoU on ADE20k. The result suggests that our deletion diagnostics strategy is more effective than other strategies for updating $\mathcal{M}$.
>
> | Method                                         |  mIoU on ADE20k  |
> | :---:                                          |  :---:           |
> | FCN baseline                                   |  36.96           |
> | FCN + IDR (back-propagation)                   |  40.35           |
> | FCN + IDR (deletion diagnostics)               |  43.61           |
>
> Second, the motivation of utilizing deletion diagnostics is to enable the network to directly utilize its objective function to examine whether one class $i$ can help recognize another class $j$. The key difference between our method and previous context modules is that we ***involve less potentially predetermined priors bias*** in the process of deletion diagnostics. Specifically, the geometric transformations of multi-scale-driven context schemes are manually set, which tends to aggregate some ineffective information and lacks of generalization.  Moreover, similarity-driven context schemes are assumed to aggregate semantically similar pixel representations, which results in ignoring other dissimilar but effective pixel representations for building co-occurrent patterns, e.g., sky pixels for airplane pixels.
>
> In a nutshell, our intervention-driven paradigm is ***more effective and have better generalization abilities*** compared to the existing similarity- / multi-scale-driven context modules. In the revised version, we will further explain this point to more clearly motivate the proposed deletion diagnostics procedure.

---

> > ### Comment · Reviewer_q3X9 · 2023-08-13
> >
> > Thanks for providing the rebuttal. Additional experiments are presented, and the authors discuss more details for this submission, I can upgrade to weak accept.
> >
> > Authors need to improve the draft a lot for the camera-ready version if this submission is accepted, by explaining the method section more clearly.

---

### Official Review · Reviewer_PhSv · 2023-07-09

**Soundness:** 3 good
**Presentation:** 2 fair
**Contribution:** 3 good
**Rating:** 4
**Confidence:** 3

**Summary:**

To further enhance the segmentation performance, the authors utilize deletion diagnostics to model pixel relationships. Specifically, to address the computational cost of pixel relation modelling, they simplify it as object relation modelling. Then, they introduce deletion diagnostics to facilitate the network in building object relationships. The experimental results qualitatively and quantitatively demonstrate the effectiveness of their proposed paradigm.

**Strengths:**

To improve the contextual information aggregation，the authors propose a novel Intervention-Driven Relation Network (IDRNet), which leverages a deletion diagnostics procedure to guide the modelling of contextual relations among different pixels. This method brings consistent performance improvements to state-of-the-art segmentation frameworks and achieves competitive results on popular benchmark datasets.

**Weaknesses:**

1. The paper lacks the presentation of additional time-consuming and memory-consuming aspects associated with the Intervention-Driven context module.
2. The paper lacks interpretability regarding how the Intervention-Driven context module improves contextual information aggregation.
3. Some minor issues: confusion in the writing, some symbols used in equations or throughout the paper are not adequately explained.

**Questions:**

1. There is confusion in the writing. For example, in the paper, IDRNet refers to an additional module that enhances contextual information aggregation. However, in Table 1, IDRNet is described as a simple FCN segmentation framework equipped with IDRNet. This inconsistency can cause confusion for readers.
2. Without using the Intervention-Driven context module, would combining the Multi-Scale-Driven and Similarity-Driven modules yield better results?
3. What is the extent of the additional time-consuming and memory-consuming aspects associated with the intervention-Driven context module?
4. In equation (8), the meaning of $R_a$ is not explained."
5. In equation (9), the meaning of $\oplus$ is not explained. Is it element-wise addition or channel-wise merging?

**Limitations:**

There is no limitation claimed in the paper. For example, the cost of the Intervention-Driven context module, particularly in terms of training time and storage, remains unknown.

---

> ### Author Rebuttal · Authors · 2023-08-09
>
> We sincerely thank you for the comments.  We hope that our responses could thoroughly address your concerns.
>
> > **Q1: Confusion of IDRNet.**
>
> We are sorry for the confused representation of IDRNet. In the revision, we will utilize ***IDRNet*** to indicate the whole segmentation framework and ***IDR*** to represent the intervention-driven context module to clear up the confusion. For example, “a simple FCN segmentation framework equipped with IDRNet (termed IDRNet)” will be revised as “a simple FCN segmentation framework equipped with intervention-driven context module (termed IDRNet)”.
>
> > **Q2: Lacks presentation of time-consuming and memory-consuming aspects of intervention-driven context module.**
>
> Thanks for your comment. Following the inference settings in OCRNet $^{[1]}$, we perform complexity comparison with other context modules on a single RTX 3090 Ti GPU and the results during network inference are reported in the below table. We will also add this table in the revision.
>
> | Context Module | Params | FLOPS |  Time    | Memory  |  mIoU on ADE20k (%)  |
> | :---:             |  :---:   |  :---:    |  :---:   | :---:    |  :---: |
> | OCR       |  15.12M  |  242.48G  |  16.58ms | 617.24M  |  42.47 |
> | ASPP      |  42.21M  |  674.47G  |  41.98ms | 976.06M  |  43.19 |
> | PPM       |  23.07M  |  309.45G  |  21.45ms | 960.63M  |  42.64 |
> | UperNet   |  34.75M  |  500.76G  |  36.51ms | 1429.18M |  43.02 |
> | ANN       |  22.42M  |  369.62G  |  26.58ms | 1445.75M |  41.75 |
> | CCNet     |  23.92M  |  397.38G  |  30.92ms | 986.28M  |  42.48 |
> | DNL       |  24.12M  |  395.25G  |  51.38ms | 2381.04M |  43.50 |
> | IDR (*ours*)      |  10.79M  |  155.89G  |  20.52ms | 365.66M  |  43.61 |
> | PPM+IDR (*ours*)  |  23.65M  |  349.23G  |  32.64ms | 1034.28M |  44.02 |
>
> It is observed that IDR module requires the least Params, FLOPS and GPU Memory,  while achieving the best mIoU on the ADE20k dataset compared with other context modules. Moreover, our IDR module is complementary to other context modules. For example, when incorporated with PPM (***PPM+IDR***), the Params, FLOPS, Time and Memory only increase by $0.58$ M, $39.78$ G, $11.19$ ms and $73.65$ M, respectively, which shows that our proposed IDR is light and efficient during network inference. Note that few parameters increase in ***PPM+IDR*** as we adopt a shared $3 \times 3$ convolution to reduce the feature channels of the backbone outputs.
>
> [1] Yuan et al. "Object-contextual representations for semantic segmentation." ECCV 2020.
>
> > **Q3: Results of combining the multi-scale-driven and similarity-driven module.**
>
> Following your suggestion, we add the results of different combinations of multi-scale-driven, similarity-driven and intervention-driven context modules.
>
> | Context Modules                                 |    mIoU on ADE20k  |
> | :---:                                                     |  :---:  |
> | multi-scale-driven (UperNet)                              |  43.02  |
> | similarity-driven (OCRNet)                                |  42.47  |
> | intervention-driven (ours)                                |  43.61  |
> | multi-scale-driven (UperNet) + similarity-driven (OCRNet) |  41.57  |
> | multi-scale-driven (UperNet) + intervention-driven (ours) |  44.84  |
> | similarity-driven (OCRNet) + intervention-driven (ours)   |  44.50  |
>
> It is observed that intervention-driven context module is complementary to multi-scale-driven and similarity-driven context modules, *i.e.*, brings 1.82% and 2.03% mIoU improvements to multi-scale-driven and similarity-driven module, respectively. On the contrary, the performance of multi-scale-driven (UperNet) + similarity-driven (OCRNet) is a little poor compared with their original performance (41.57% *v.s.* 43.02% and 42.47%). The reason may be that such a combination is difficult to train or may easily lead to overfitting.
>
> > **Q4: How the Intervention-Driven context module improves contextual information aggregation.**
>
> The motivation of intervention-driven context module is to enable the network to _**directly utilize its objective function to examine whether one class $i$ can help recognize another class $j$**_. Specifically, if the removed contextual pixels of class $i$ is helpful for recognizing the pixels of class $j$, the loss values of class $j$ should increase and such increased value can be used to reflect the relation between class $i$ and $j$ in $\mathcal{M}$. Consequently, class $j$ can aggregate more effective contextual information from class $i$ according to the estimated relation value to facilitate network optimization and improve segmentation accuracy.
>
> The key difference between our module and previous context modules is that we involve ***less potentially predetermined priors bias*** during pixel context modeling. Particularly, the geometric transformations of multi-scale-driven context schemes are manually set, which tends to aggregate some ineffective information and lacks of generalization. Similarity-driven context schemes are assumed to aggregate semantically similar pixel representations, which results in ignoring other dissimilar but effective pixel representations for building co-occurrent patterns, *e.g.*, sky pixels for airplane pixels.
>
> > **Q5: Some minor issues.**
>
> - $R_a \in \mathbb{R}^{Z \times \frac{H}{8} \times \frac{W}{8}}$ is the feature representations by re-arranging $R_{esl}$. Specifically, we initialize $R_a$ as a zero matrix and fill it with the representations in $R_{esl}$ according to the pseudo labels in each pixel position. For example, if the labels in the input image $I$ are {$0, 1, 5, 7$}, $R_{esl}$ will be {$R_{esl}^{cls=0}, R_{esl}^{cls=1}, R_{esl}^{cls=5}, R_{esl}^{cls=7}$} in Eq (6). And if class id of $R_{a, [\*, 0, 0]}$ is 5, we will set $R_{a, [\*, 0, 0]} = R_{esl}^{cls=5}$ in Eq (8).
>
> - $\oplus$ indicates a concatenation operation as explained after Eq (3). In the revision, we will further explain it after Eq (9) to easy understanding.

---

### Decision · Program_Chairs · 2023-09-21

**Decision:**

Accept (poster)

**Comment:**

This paper received varying scores, with three reviewers recommending acceptance and one reviewer suggesting rejection. The authors effectively addressed the reviewers' comments during the rebuttal phase. This paper presents a novel approach aimed at enhancing contextual information aggregation in the field of image segmentation.  This paper introduces the deletion diagnostics mechanism to guide the modeling of contextual relationships among various visual pixels. The authors rigorously examine the impact of different contextual design choices through a series of ablation experiments, which the AC believe is inspiring to the segmentation community. The evaluations across five different datasets are notably impressive and solid. Furthermore, the authors effectively addressed the reviewers' comments during the rebuttal phase and the majority of reviewers have expressed a positive opinion of this paper. Considering the overall assessment and the improvements made by the authors, the AC has decided to accept this paper, despite its averaged rating is lower than the acceptance bar.